# Physiological Responder Profiles and Fatigue Dynamics in Prolonged Cycling

**DOI:** 10.3390/jfmk10040472

**Published:** 2025-12-08

**Authors:** Adrian Odriozola, Cristina Tirnauca, Francesc Corbi, Adriana González, Jesús Álvarez-Herms

**Affiliations:** 1Hologenomiks Research Group, Department of Genetics, Physical Anthropology and Animal Physiology, University of the Basque Country (UPV/EHU), 48940 Leioa, Spain; adriana.gonzalez@ehu.eus; 2Institut Nacional d’Educació Física de Catalunya (INEFC), Centre de Lleida, Universitat de Lleida (UdL), 25003 Lleida, Spain; jesusah80@gmail.com; 3Departamento de Matemáticas, Estadística y Computación, Universidad de Cantabria, 39005 Santander, Spain; cristina.tirnauca@unican.es; 4Department of Clinical Sciences, Faculty of Medicine and Health Sciences, University of Barcelona, 08907 L’Hospitalet de Llobregat, Spain; f@corbi.neoma.org; 5Phymo^®^ Lab, Physiology, and Molecular Laboratory, 40170 Collado Hermoso, Spain

**Keywords:** bicycling, physical endurance, muscle fatigue, biomarkers, phenotype, athletic performance

## Abstract

**Objectives**: To characterise multidomain physiological responses to a maximal cycling effort and identify consistent physiological responder profiles. A secondary objective was to compare professionals and amateurs and assess the practical value of these profiles for personalised monitoring and performance management. **Methods**: This observational study included 22 trained male cyclists (10 professionals, 12 competitive amateurs; age 27.6 ± 6.4 years; height 177.3 ± 5.5 cm; weight 65.5 ± 4.1 kg). Participants performed a maximal 20-min functional threshold power (FTP) test and complementary assessments (Bosco jump tests, blood pressure, heart rate, lactate, glycaemia, creatine kinase, albuminuria) across three time points (baseline, immediately post-FTP, and 24 h post-FTP). Statistical analyses included *t*-tests, ANOVA, and Spearman correlations, for recovery dynamics, with significance set at *p* < 0.05. **Results**: Professionals exhibited significantly higher FTP (5.5 ± 0.3 vs. 4.3 ± 0.4 W/kg, *p* < 0.001), greater post-exercise lactate (13.8 ± 1.6 vs. 11.2 ± 1.4 mmol/L, *p* < 0.01) and higher CK 24-h responses (412 ± 86 vs. 291 ± 74 U/L, *p* < 0.05). Cardiovascular and metabolic recovery slopes were faster in professionals (*p* < 0.05). Despite lower baseline jump values, professionals showed reduced neuromuscular fatigue (SJ post/pre = 0.94 ± 0.04 vs. 0.88 ± 0.05, *p* < 0.05). FTP correlated strongly with 5-min all-out power (*r* = 0.76, *p* < 0.01) and Wingate mean power (*r* = 0.75, *p* < 0.01). Eight responder profiles emerged across four physiological domains, with professionals predominantly showing multi-domain adaptation patterns. Although additional variables, such as elevated albuminuria and altered Elasticity Index (EI), provide insight into renal and neuromechanical stress responses, they were excluded from the final profiling due to limited practical interpretability. **Conclusions**: Fatigue and recovery in prolonged cycling show substantial interindividual variability across neuromuscular, metabolic, cardiovascular, and biochemical domains. Professional cyclists display faster recovery and more frequent multidomain responder profiles. The four-variable model (FTP, lactate, CK, SJ post/pre) enables clear identification of physiological responder types and offers a practical, integrative framework for personalised monitoring and performance management.

## 1. Introduction

Fatigue in cycling performance could be described as the inability to maintain a required or expected power output, leading to a performance loss in a given task [1]. In this regard, the decline in exercise performance has been associated with the impairment of different systemic physiological thresholds affected by the intensity and volume of exercise, mainly related to metabolism [2].

Cycling fatigue arises from the interaction of peripheral, metabolic, cardiovascular, and central regulatory mechanisms. High-intensity continuous pedalling increases glycolytic flux, leading to the accumulation of H^+^ and inorganic phosphate, impaired calcium handling, and reduced excitation–contraction efficiency [3]. Cardiovascular and respiratory constraints limit oxygen delivery during sustained high-intensity effort, reduce arterial oxygenation, and compromise aerobic ATP resynthesis [4]. Metabolic stress includes progressive carbohydrate depletion, thermal strain, and redox imbalance, further reducing the capacity to sustain power output during prolonged exertion [5]. Simultaneously, afferent feedback from group III–IV muscle receptors increases central motor inhibition [6].

Endurance exercise increases multisystemic disruption as longer and more intense exercise is required, altering metabolic stress, including acidosis, thermal load, and oxygen availability with impaired physical/perceptive performance and neuromuscular efficiency [7,8].

Multisystemic fatigue collapses the capacity to provide energy efficiently to the muscles in a dose-dependent manner, leading to increased inhibitory feedback to central motor drive [2]. Previous studies proposed complex system models to explain fatigue in sports where afferent somato-sensorial feedback information change our unconscious perception of fatigue [9,10].

On the other hand, the conscious rating of perceived effort (RPE) integrates physiological and psychological inputs to reflect the subjective experience of exertion during exercise, playing a key role in the central nervous system’s regulation of neural responses to organs and tissues [10,11]. Based on previous experiences and anticipatory neural models, allostasis describes how peripheral and central inputs [12,13] and expected exercise duration are regulated [14]. According to Swart et al. [14], the perceptive cues during submaximal cycling exercise play a critical role in regulating exercise intensity because they alter the behaviour of voluntary control of physical performance.

Current strategies to mitigate fatigue in cycling rely on optimising pacing, metabolic efficiency, neuromuscular resilience, and environmental management. Evidence shows that even pacing or slight negative pacing reduces early glycolytic depletion and delays neuromuscular fatigue during prolonged efforts [15]. Nutritional interventions such as adequate carbohydrate availability, sodium–fluid balance, and caffeine intake help maintain metabolic homeostasis and perceptual regulation during high-intensity cycling [16,17,18]. Strength and plyometric training improve cycling economy and fatigue resistance by enhancing neuromuscular recruitment and muscle–tendon efficiency [19,20]. Heat acclimation and cooling strategies attenuate thermal load and preserve cardiovascular stability, especially in long or intense sessions [21]. Together, these approaches highlight that fatigue mitigation in cycling requires an integrative strategy combining metabolic, neuromuscular, perceptual, and environmental interventions.

Recent studies identify consistent physiological differences between professional and amateur cyclists across metabolic, neuromuscular, and perceptual areas. Professional cyclists generally exhibit higher lactate turnover rates, greater mitochondrial density, improved buffering, and better glycolytic–oxidative coupling during high-intensity efforts, supporting superior FTP and sustained power outputs [22]. Neuromuscularly, professionals demonstrate more efficient muscle–tendon behaviour, increased fatigue resistance, and faster recovery kinetics, reflecting long-term exposure to high training volumes and cycling-specific mechanical loads [7]. Perceptually, elite cyclists maintain more stable RPE regulation and more precise interoceptive control during intense efforts, allowing for more effective pacing and tighter regulation of homeostatic disturbance [14]. This evidence underscores the importance of comparing professionals and amateurs when studying physiological responder profiles.

Professional cycling involves reaching the maximum physiological limits of systemic fatigue, with high levels of RPE during specific mechanical cycling movements [23]. Hence, cycling training is characterised by accumulating high loads and intensities to stimulate physiological levels [22].

Traditionally, measuring the maximum oxygen uptake (VO_2_max) and its fractional use during competition has been related to endurance performance, categorising levels for athletes and cyclists [24]. However, among well-trained athletes with similar VO_2_max values, other psychobiological and mechanical factors are even more determinant in explaining inter-individual differences in riding performance [24,25,26,27]. One of them is functional threshold power (FTP), which directly reports the highest constant power mean output during pedalling at a fixed time, ranging from 20 to 60 min [28,29,30,31]. FTP has been demonstrated to be a highly reliable, applicable, and practical parameter to measure on the road, thanks to the information provided by power meters installed on bikes [28,29,30,31,32,33,34]. Also, FTP has been validated in laboratory environment through cardiovascular [35] and respiratory effort tests [29]. In addition, previous studies found that FTP is a useful tool to classify cyclists according to their level [29], establishing a minimum power mean value between 6.0–6.5 W/kg, as a minimum requirement to be a professional cyclist [36].

On the other hand, the decreased neuromuscular performance is an evident sign of neural fatigue in cyclists [37]. In this context, Bosco’s protocol tests (BT) has been described as an efficient tool for measuring neuromuscular performance [38,39,40] and fatigue [41,42,43], though a series of jump tests for the assessment of leg muscular mechanics and power. We previously showed that BT performance correlates directly with a higher performance of BMX riders, probably thanks to their neuromuscular profile [44] and malleability [45]. Likewise, most studies have described how BT performance could be an easy, efficient and non-invasive tool, especially in outdoor environments, for detecting central and peripheral fatigue, thanks to its capacity to detect neuromuscular recruitment fatigue [41,42,43]. Unfortunately, to the best of the author’s knowledge, applying BT in cycling science as a tool to describe premature peripheral fatigue is a not previously used strategy, especially when it is tested in parallel with other physiological variables.

For these reasons, we hypothesise that combining different tests, both specific to cycling power (maximal FTP) and more general for neuromuscular power (BT), offers a practical and cost-effective strategy to characterise physiological responses to prolonged high-intensity exercise in trained cyclists. We further claim that these responses vary substantially between individuals and can be clustered into distinct physiological responder profiles, reflecting coordinated adaptations or vulnerabilities across cardiovascular, metabolic, muscular, renal, and neuromuscular systems. These profiles are expected to influence fatigue, recovery, and performance outcomes.

Therefore, the main objective of this study was to characterise the multidomain physiological responses triggered by a maximal cycling effort and to determine whether these responses can be organised into consistent physiological responder profiles. A secondary goal was to compare these profiles between professional and amateur cyclists and to evaluate their practical significance for personalised monitoring, training management, and performance improvement. Evidence suggests that individualised interventions based on integrated physiological phenotyping may enhance adaptation, reduce injury risk, and support long-term athlete health and performance [46,47].

## 2. Materials and Methods

### 2.1. Study Design and Testing Environment

This study used a cross-sectional, repeated-measures design and was conducted during the competitive season. All data were collected at the INEFC–University of Lleida (Spain) and on the cyclists’ own bicycles equipped with a calibrated Powertap G3 hub. Testing sessions were performed under controlled environmental conditions (temperature 20–22 °C; humidity 40–60%).

### 2.2. Experimental Approach to the Problem

The performance exhibited by the participants, along with their main characteristics, is indicated in Table 1. Cyclists participated on three different days for the measurements during the season (see Figure 1 and Table 2). On day 1, anthropometric (weight and height), basal physiological measures, BT jump tests and “all out” 30″ Wingate and 5′cycling tests were done. During day 2, 72 h after, a 20′ FTP test was carried out, and the corresponding physiological measures of recovery, BT jump tests and RPE were carried out. On day 3, 24 h after the FTP test, physiological measures of recovery, BT jump tests and the rating of perceived fatigue (RPF) were recorded.

### 2.3. Participants

#### 2.3.1. Participant Characteristics

Twenty-two competitive male cyclists participated in the study (10 professionals and 12 competitive amateurs). All participants completed the three testing sessions and provided written informed consent. The study adhered to the ethical principles of the Declaration of Helsinki [48], the Spanish Biomedical Research Law 14/2007, and the European Regulation (EU) 2016/679 on data protection (GDPR). Ethical approval was granted by the Human Research Ethics Committee of the University of the Basque Country (M10 2021 191). All tests were performed during the seasonal phase, and cyclists followed their regular training regimen. Participants refrained from physical training for three days before and after the FTP test.

#### 2.3.2. Inclusion and Exclusion Criteria

Cyclists were eligible if they were male competitive athletes aged 18–40 years, free of injury during the previous three months, and actively training and competing during the season. Only male participants were included to minimise variability associated with sex-specific physiological and hormonal differences. Exclusion criteria included cardiovascular, metabolic, endocrine, renal, or neuromuscular disorders; the use of medications affecting performance or recovery; and failure to complete all testing sessions. A study flowchart illustrating the participant pathway is presented in Figure 2.

#### 2.3.3. Classification of Performance Level

Professional cyclists were defined as athletes competing in elite-level teams with a structured national or international competition calendar and systematic training support typical of professional practice. Amateur cyclists were those participating in regional or national events without a professional contract or elite-team structure.

### 2.4. Procedures

#### 2.4.1. Physiological Measurements

Body mass was measured with a mechanical column scale (Seca 711, Seca GmbH, Hamburg, Germany; accuracy 0.1 kg), and height was measured using the attached stadiometer (Seca 220, Seca GmbH, Hamburg, Germany; accuracy 0.1 cm). Body mass index (BMI) was calculated as weight (kg) divided by height squared (m^2^). Physiological measures included basal heart rate (seated and standing) with a pulsometer model Polar Rs410 (Polar electro^®^, Kempele, Finland), blood pressure (Omron M6; © OMRON Healthcare, Kyoto, Japan), lactate (Lactate Plus, Nova Biomedical, Waltham, MA, USA, EE. UU) and glycaemia (Accu-Chek^®^ Guide, Roche Diagnostics, Mannheim, Germany). In addition, on days 1 and 3, samples of creatine kinase (CK) (mobile photometer Vario II DP 310) and albumin (Albumin RIA 100; Pharmacia AB, Uppsala, Sweden, urine test strips) were measured. Albumin RIA 100 is based on purified human serum albumin as a standard, ^125^I-labelled human serum albumin as a tracer, and a specific antiserum to human serum albumin. It has a lower sensitivity limit of 0.38 mg/L and has been widely used in athletes [49]. Its analytical features have been widely described previously [50]. The samples were collected according to the manufacturer’s instructions [51].

#### 2.4.2. Bosco Test Battery Test

Bosco’s battery tests included squat jumps (SJ), countermovement jumps (CMJ), and continuous countermovement jumps for 30 s (CMJ30). All measures were calculated with a contact platform model, Chronojump (Boscosystems^®^, Barcelona, Spain) [52].

#### 2.4.3. All-Out Cycling Tests

Cycling tests of Wingate and 5 min all-out” were performed in a cycle ergometer model Wattbike (Wattbike Ltd.; Leeds, UK), previously used in cyclist tests [53]. To measure the cycling power performance during the FTP test, the rider’s bicycles were equipped with the same interchangeable mobile power meter positioned in the rear wheel (Powertap professional power meter, model G3 Hub. This device model has been previously demonstrated to have high reliability, sensitivity, and reproducibility of measures compared to other validated models [54]. To ensure accurate measurements, a static calibration procedure was performed before the study, as per the manufacturer’s technical instructions (further details available at https://www.sram.com/en/quarq, accessed on 23 November 2025).

The FTP test consisted of a maximal test performance over 20 min, performed at all-out effort. During the FTP test, two investigators monitored participants’ time in both seated/standing positions. Finishing the FTP, cyclists stopped and performed RPE and repeated physiological measures (heart rate, blood pressure, glycaemia and lactate) during 0, 3, 5, 10, and 20 min. During the measurements, cyclists rested in a seated position and refrained from eating or drinking anything. After 20 min of recovery, Bosco’s jump tests, which were performed on day one, were repeated, and the RFP was carried out.

#### 2.4.4. Statistical Analysis

Data normality was assessed using the Shapiro–Wilk test. For each considered variable, the mean, standard deviation were presented. The neuromuscular fatigue rate was calculated by considering the percentage of jump capacity after the FTP test (immediately and 24 h after), as well as the jump capacity before the FTP test (BTpost/BTpre × 100), and was expressed as a percentage [55]. The Elasticity index (EI) was calculated using the following formula: EI = (CMJ − SJ)/SJ × 100.

2-tailed *t*-tests analysed differences between professionals’ and amateurs’ groups. Spearman correlation coefficients (r) were analysed from the participants’ data. A two-way mixed ANOVA with repeated measures (ANOVA group × time) was used to evaluate the effects of time and group (professional vs. amateur). Time was considered as pre/post/24 h for the variables SJ, CMJ, CMJ30″, EI, and as 0, 3, 5, 10, and 20 min for HR (heart rate), SBP (systolic blood pressure), DBP (diastolic blood pressure), glycaemia, and lactate. In the presence of significant effects (*p* < 0.05), the analysis was complemented with the Holm-Šídák post hoc test. The recovery dynamics after the FTP test for HR, SBP, DBP, lactate, and glycaemia were also calculated using linear and quadratic regression. The differences in the recovery dynamics of professional and amateur groups were assessed by examining the linear, quadratic, and cubic regression slopes. Statistical significance was assumed when *p* < 0.05 and a non-statistically significant tendency when 0.05 < *p* < 0.06.

A heatmap derived from post-exercise variables was carried out to analyse physiological responder profiles. All data processing and visualisation were performed using Python 3.10 with the libraries Pandas (version 1.5.3), NumPy (version 1.23.5), Seaborn (version 0.12.2), and Matplotlib (version 3.5.3). Each athlete was classified as a responder in a given domain if the following thresholds were met for: FTP ≥ 5.5 W/kg (high endurance capacity, cardiovascular-metabolic domain [56]); lactate > 13 mmol/L (glycolytic activation [57]); serum creatine kinase (CK) levels 24 h post-exercise > 400 U/L (muscle membrane stress or adaptation [58]), and the percentage change in squat jump height (SJ post/pre%) ≥ 0.90 (neuromuscular function recovery [59]). Z-scores were computed to standardise interindividual variability, and composite phenotypes were defined based on the exact combination of domains where each cyclist met the responder criteria. Visualisation was performed using hierarchical clustering with Euclidean distance and complete linkage, and responder categories were defined a priori based on physiologically relevant thresholds. Additional heatmaps were constructed using 11 post-exercise variables, but were excluded from the final analysis due to high dimensional overlap, lack of dominant patterns, and reduced practical interpretability.

All the statistical calculations were done using GraphPad Prism version 10.0.0 for Windows, GraphPad Software, Boston, MA, USA.

## 3. Results

Table 1 summarises participants’ main features and cycling performance for Wingate, 5, and 20-min FTP. The amateur and Professional groups showed similar characteristics in terms of age, weight, height, and standing time during the 20-min FTP test. In contrast, professionals exhibited lower BMIs and achieved significantly higher cycling performance.

Table 2 presents the significant correlations between professional status, represented in green and red, indicating positive and negative relationships, respectively. Additionally, this table details the association between better FTP test performance and the other analysed variables. This allows for the simultaneous visualisation of features related to professional status and/or to better performance in all participants, not only professionals or amateurs. Hence, features associated with both status and performance in all participants include: on day 1, positively every all-out cycling test performance, RPE and CK levels, and negatively SJ, CMJ, HR and DBP basal values; on day 2, after the FTP test, positively SJ, CMJ, glycaemia and lactate values, and higher increase and decrease in blood pressure; and on day 3, higher SJ and CK values.

The main correlations not associated with professional status but with FTP test performance include: on day 1, a lower CMJ 30′ and EI only in professionals; on day 2, higher CMJ 30′ and higher HR changes in all participants and higher glycaemia during 5–10 min after FTP test only in amateurs. Lactate values at minute ten after the FTP test were the only feature clearly showing a contradictory association between professionals and amateurs. On day 3, a lower EI was associated with better FTP test performance.

To facilitate understanding of the biology of cycling performance, the interactions between the analysed dynamic physiological variables are described in Table 3. Most pair interactions between variables showed significant positive correlations, suggesting that the magnitude of their changes was interrelated. The exception to this rule was lactate, which did not show a significant association with HR and exhibited a negative correlation with DBP. To summarise the dynamics of continuous variables, Figure 3A visualises the behaviour of variables immediately after the FTP test across all participants, specifically highlighting differences between professionals and amateurs in glycaemia-lactate interactions, as shown in Figure 3B.

To provide an integrated overview of how physiological domains relate to each other across the testing timeline, Figure 4 summarises the significant between-group differences and the network of correlations observed during the study. This representation allows visual inspection of three complementary layers of the data: (i) basal and post-FTP differences between professionals and amateurs, (ii) the structure of significant associations among metabolic, cardiovascular and neuromuscular variables, and (iii) the temporal evolution of key post-exercise variables (SBP, DBP, lactate and glycaemia) at 0–20 min. Together, these elements highlight the multidomain nature of the physiological response to the FTP test and illustrate how group-level differences emerge from coordinated interactions among cardiovascular, metabolic, neuromuscular, and perceptual markers. For all parametric variables, the significance of the differences between professional and amateur cyclists was confirmed by both Student’s *t*-test and the mixed ANOVA model, indicating that the group effects were statistically robust.

The analysis of post-exercise responses was conducted to investigate the individual physiological profiles of the cyclists. Using the four-variable model (FTP W/kg, post-exercise lactate, CK 24 h, and SJ post/pre), cyclists were categorised into eight different responder profiles and the non-responder profile, reflecting predominant adaptation patterns in cardiovascular–metabolic, glycolytic, muscular, or neuromuscular systems (Figure 5). This classification revealed a heterogeneous distribution across professional and amateur athletes, with most professionals (5/7) exhibiting dual or triple-domain responder phenotypes more frequently. In contrast, amateurs were more likely to present isolated or non-responder patterns. Additionally, broader models were explored, including all the post-exercise variables, but they were discarded due to their limited practical interpretability.

## 4. Discussion

The present study evaluated neuromuscular performance (BT) and its relationship with physiological dynamics during a high-intensity endurance stimulus (FTP test) in amateur and professional cyclists. FTP performance was higher in professionals, who also achieved more extreme physiological values and faster recovery. FTP results aligning with professional benchmarks, higher than 6.0 W/kg in incremental tests [36]. FTP differed significantly between groups and is easier to measure in both training and competition than VO_2_max [24]. FTP correlated strongly with 5-min all-out, Wingate mean power and peak power, supporting its value as an integrative performance metric [60]. While Wingate peak power can help exclude non-professionals, it cannot classify long-endurance capacity due to physiological differences between short and prolonged high-intensity efforts, which disrupt metabolic homeostasis and induce peripheral fatigue [2,9,10,61,62]. Consistent with previous research on endurance cyclists, professionals showed a significantly lower BMI than amateurs. This anthropometric pattern aligns with their higher relative power outputs and superior performance in FTP, Wingate, and 5-min tests, suggesting a favourable body composition profile typical of high-level road cycling [63,64].

### 4.1. FTP Test as a Cyclist-Specific Physiological Performance Indicator

Taken together, the correlation patterns observed between FTP and cardiometabolic variables (particularly lactate, blood pressure, and glycaemic responses) highlight the importance of lactate turnover and cardiovascular regulation as central determinants of sustained high-intensity cycling performance. This aligns with previous work showing that metabolic flexibility and vascular responsiveness critically shape endurance capacity [65].

### 4.2. FTP vs. BT

An innovative point of this study was to assess how a maximal endurance cycling effort (FTP test) influenced neuromuscular function, measured via BT. We found no previous studies that quantified the effect of individual physiological stress on jump performance.

Lower BT performance before the FTP in professionals likely reflects adaptations to cycling-specific stimuli, as professionals train 16–25 h per week [66], which promotes a plateau in metabolic homeostasis at a relative VO2 max but reduces recruitment in non-specific explosive tasks.

Due to reduced neuromuscular fatigue exhibited by professionals after the FTP test, we hypothesise that neuromuscular potential loss during endurance activity relates to reduced firing frequency and contractile tension [67], with peripheral muscle power generation depending on efficient recruitment when metabolic homeostasis is altered [8,68,69]. In other words, endurance-trained athletes maintain contractile function more effectively under metabolic strain.

### 4.3. FTP vs. Elasticity Index (EI)

The EI has traditionally been used as a reliable measure of the stretch–shortening cycle (SSC) and neuromechanical performance in athletes [70,71]. It has also been applied to assess explosive strength and elastic tissue function during jumping tasks [72,73]. However, in the present study, higher pre-FTP EI was significantly associated with greater relative fatigue and incomplete 24-h recovery across jump modalities [74].

A subgroup of four cyclists (predominantly amateurs) with EI > 30% showed more pronounced fatigue (mean SJ decline −9.1 ± 1.8%) and delayed neuromuscular recovery (BT recovery ratios < 0.90) alongside slower HR recovery post-FTP.

These data indicate that elevated EI, often advantageous in explosive tasks [70,71,72,73], may reflect reduced neuromuscular efficiency under metabolic stress in endurance cycling, where excessive compliance can impair rapid force transmission during sustained effort [75]. This pattern supports a mechanical–muscular responder phenotype: high EI with suboptimal stiffness regulation compromises energy transfer when homeostasis is perturbed, magnifying fatigue and delaying recovery [74]. Practically, such athletes may benefit from neuromuscular work aimed at stiffness control and eccentric force transmission, complementing endurance-specific conditioning.

### 4.4. FTP vs. BT and Cycling Position

Time spent in seated vs. standing positions during the FTP test showed no association with post-exercise BT variation (Table 1). Several studies report no significant effect of position on ventilatory [76,77], cardiovascular [78], or metabolic parameters [79] at maximal workloads. Conversely, others observed differences in ventilatory and metabolic responses at submaximal intensities [80,81]. In our cohort, greater standing time correlated negatively with absolute jump capacity pre-, post-, and 24 h post-FTP, suggesting that prolonged standing, particularly during climbs, may be related to reduced non-specific neuromuscular power, independent of FTP.

### 4.5. Performance and Physiological Recovery Dynamics in Professional vs. Amateur

The target physiological variables of all participants change dramatically during and immediately after the FTP test (Figure 3). Overall, the professional group reported significantly more extreme basal physiological values and higher extreme physiological values achieved immediately after the FTP test. These higher physiological limits reached by professionals may be due, at least in part, to specific adaptive responses induced by the repeated stimulus of endurance activities over the long term [66].

In the case of maximal blood pressure levels reached during maximal endurance exercise, previous studies have demonstrated marked blood pressure [82]. In the present study, elevated blood pressure was significantly correlated with FTP test performance in all participants and was more pronounced in professionals than in amateurs. After the FTP test, professionals showed significantly higher systolic and diastolic blood pressure and faster recovery dynamics. Only two professional cyclists exceeded the 210 mmHg systolic threshold, previously associated with greater vascular elasticity and left ventricular hypertrophy [83]. In coherence, these two individuals also displayed the maximum heart rate and DBP values and the fastest recovery dynamics from all participants. This pattern reflects a cardiovascular adaptive responder phenotype, characterised by the ability to reach high systolic pressures during exertion and recover rapidly thereafter, as well as a high performance. These traits are linked to enhanced vascular complications and cardiac output reserve; they may serve as practical biomarkers for identifying athletes with superior cardiovascular conditioning and recovery capacity [83,84]. Cardiac remodelling is more intense in cycling than in other sports practices, and it is associated with a supernormal pattern of left ventricular diastolic function compared to non-athletes [84].

In the present study, a significant increase in albuminuria was observed 24 h after the FTP test, with a mean of 22.1 ± 2.7 mg/24 h, exceeding values typically reported in healthy individuals after maximal exertion. This proteinuria aligns with previous studies describing transient post-exercise increases in glomerular permeability due to intense physical stress [49,85,86,87]. Notably, although Kramer et al. [85] did not report statistically significant increases in albuminuria following a maximal cycling test in healthy participants; their mean values of 10.3 ± 0.9 mg/24 h remained well below those in our cohort. This discrepancy likely reflects differences in both exercise intensity and training status. Indeed, the FTP test used in our protocol produced substantially higher lactate concentrations, which may have induced a more pronounced systemic acidosis and a glomerular stress response. Accordingly, albuminuria can be interpreted within the broader multidomain responder framework as an indicator of systemic metabolic strain.

A closer analysis revealed that the three cyclists exhibiting the highest albuminuria values (≥30 mg/dL) also presented the greatest lactate accumulation (>14.5 mmol/L), the highest creatine kinase elevation at 24 h, and superior performance in the FTP test (>5.5 W/kg). These individuals further demonstrated faster glycaemia clearance and accelerated recovery of heart rate and blood pressure following exertion. The physiological coherence of this response suggests a coordinated systemic adaptation to intense metabolic stress.

From a mechanistic standpoint, the accumulation of lactate and hydrogen ions during high-intensity exercise is known to alter the permeability of the glomerular capillary wall and impair tubular reabsorption capacity, thereby promoting transient protein leakage into the urine [49,87,88,89]. This relationship between glycolytic engagement, acidosis, and renal protein handling has been proposed as a hallmark of the acute renal response to metabolic overload in trained athletes [48,49,90,91].

The group of responses, characterised by elevated lactate, proteinuria, CK, and rapid systemic recovery, defines a metabolic–renal responder phenotype. It shows high anaerobic capacity, resilience to acidosis, and acute renal adaptation. Identifying such a phenotype may have practical applications in performance diagnostics and recovery monitoring [89,91].

Lactate and glycaemia levels were also significantly correlated, likely due to glucose being the preferred fuel for muscle and heart metabolism during FTP test-type stimuli, producing lactate via anaerobic glucolysis [92]. Likewise, glycaemic and lactate levels correlated with FTP test performance significantly in all participants, and professionals showed both higher glycaemia values and faster recovery dynamics immediately after the FTP test than amateurs (Table 2), probably due to the higher glucose availability expected by blunted catecholamine response to exercise in maximal stress response in more adapted athletes [93]. Overall, professionals exhibit greater muscle mass and are better adapted to anaerobic glycolysis, conferring a metabolic advantage in their response to high-intensity stimuli compared to amateurs.

The analysis of CK at rest and after exercise has been described as an essential tool for coaches and clinicians to assess muscle function following exercise [94]. The present study found a significant increase in CK levels 24 h after the FTP test (Table 2), indicating high interindividual variability, which ranged from 140 to 520 U/L. Professionals showed significantly higher CK values before exercise and greater increases in response to the FTP test than amateurs, in contrast to previous studies, which suggested a negative correlation between CK levels and physical training status. In those cases, after undertaking the same physical exercise test, the CK levels of athletes were lower than those recorded in matched healthy control participants [95,96].

Notably, the three cyclists with the highest CK increase (>400 U/L) also had the highest post-exercise lactate levels and RPE scores above 9, and incomplete jump recovery at 24 h. These athletes showed no clear difference in baseline CK, suggesting that the observed elevations are not due to chronic overload but acute muscle membrane disruption and insufficient recovery capacity. CK responses have been previously linked to eccentric load, training history, and individual susceptibility [94,95,96,97]. In our data, high CK responders may represent a subgroup with lower resilience to sustained high-intensity efforts, despite comparable power outputs. This profile could be informative for personalised training load management and post-race recovery strategies, especially when corroborated by jump test and subjective fatigue indicators.

The perception of fatigue and effort during exercise is a subjective sensation processed by the brain and based on previous experiences, so it is a combination of peripheral and central inputs [12,13]. Being RPE the central mechanism by which the CNS orchestrates a physiological response based mainly on the individual’s experience regarding the previous stimuli [14], it exerts a vital role in regulating voluntary control of physical performance during intense cycling exercise [14]. Hence, professionals reported higher RPE values immediately after the FTP test, RPE was negatively correlated to the percentage of neuromuscular fatigue immediately after FTP, and positively to post-FTP lactate (Table 2). This coherence suggests a perceptive-effort responder profile, reflecting accurate interoception and fatigue anticipation. This profile may offer pacing advantages in time trials, as described in psychobiological models of endurance regulation [98,99].

Our results suggest a clear relationship between greater tolerance to maximum fatigue and better cycling performance. The present results correlate with those of Swart et al., who described the importance of perceptual cues in regulating exercise intensity by influencing the voluntary control of physical performance [14]. In this regard, the results suggest that professionals have better control over perceptual sensations of effort than amateurs, allowing for more effective regulation of central motor drive in response to a more dramatic alteration of the homeostatic range [2].

### 4.6. Individual Physiological Responder Profiles

One of this study’s most actionable findings was the identification of distinct individual physiological responder profiles in endurance cyclists, as shown in a heatmap (Figure 5) [100]. Using a four-variable model (FTP, post-exercise lactate, CK at 24 h, and SJ post/pre-recovery ratio), we classified cyclists into interpretable categories, including full responders, metabolic-dominant, muscular responders, and neuromuscular-limited.

The responder thresholds were selected based on established physiological criteria and prior evidence describing domain-specific performance boundaries in trained cyclists. An FTP ≥ 5.5 W·kg^−1^ reflects the lower bound typically reported for professional-level endurance capacity and corresponds to the power–duration domain where cardiovascular–metabolic integration becomes a primary limiter [56]. Lactate concentrations > 13 mmol·L^−1^ were chosen because they indicate a high glycolytic activation state, characterised by rapid anaerobic flux, marked H^+^ accumulation, and substantial perturbation of acid–base balance, responses consistently observed during maximal or near-maximal cycling efforts [57]. A 24-h post-exercise CK value > 400 U·L^−1^ has been widely used to differentiate normal exercise-induced muscle membrane stress from more pronounced sarcolemmal disruption, capturing interindividual variability in mechanical load tolerance and repair kinetics [58]. Finally, an SJ post/pre ratio ≥ 0.90 is supported by research indicating that decrements higher than 10% represent meaningful neuromuscular fatigue and incomplete recovery of stretch–shortening cycle function following high-intensity endurance work [59]. Together, these thresholds provide mechanistically grounded and practically interpretable markers spanning cardiovascular–metabolic, glycolytic, muscular, and neuromuscular domains, enabling multidimensional profiling of individual physiological response patterns.

Profiles aligned with performance level: most professionals exhibited full or dual-domain responses, while amateurs more often showed single-domain types. Full responders demonstrated strong adaptation across all domains, suggesting superior systemic tolerance to high-intensity effort and efficient recovery dynamics [101]. Non-responders failed to meet thresholds, exhibiting low FTP, low lactate, modest CK, and incomplete neuromuscular recovery, possibly reflecting insufficient training, subclinical fatigue, or low phenotypic plasticity [101,102].

Intermediate profiles also conveyed meaningful information. Muscular responders showed elevated CK without concurrent glycolytic or neuromuscular activation, suggesting disproportionate peripheral strain with insufficient systemic engagement [98]. Glycolytic responders exhibited high FTP and lactate, but impaired neuromuscular recovery, indicating rapid energy mobilisation at the cost of motor integrity [57]. Such imbalances may warrant targeted interventions, such as improved recovery for metabolic responders or neuromuscular strengthening for muscular responders.

These classifications offer translational potential: training and monitoring could be tailored to address domain-specific limitations, particularly in athletes presenting imbalanced profiles. Furthermore, early identification of non-responders or maladaptive responders could support preventive strategies against overreaching or chronic fatigue [98,101,103,104].

Finally, profiling athletes by physiological response domains, rather than solely based on absolute performance, favours the development of individualised training and recovery strategies, especially in high-performance contexts where marginal adaptation gains are crucial [46,102].

### 4.7. Limitations and Future Directions

Despite the strengths of a multidomain, athlete-centred physiological characterisation, several methodological considerations must be acknowledged to contextualise the current findings and guide future research. First, the sample size was small (*n* = 22) and included only trained male cyclists. Although using a homogeneous male group reduces physiological variability, this choice necessarily limits generalisability and prevents analysis of sex-specific responses in neuromuscular, metabolic, and renal areas. Sex-related differences in fatigability and recovery are well documented. For instance, females may exhibit greater fatigue resistance in some situations [105], and their exclusion can significantly limit the model’s external validity. Second, the self-paced nature of the 20-min FTP test, designed to reflect ecological performance, may introduce individual variability in pacing strategy and motivation [106]. While this enhances applicability for real-world cyclists, it reduces mechanistic control and does not study the specific contributions of certain physiological domains.

Third, although pre-test nutrition, hydration status, recent training load, and sleep are well-established modulators of metabolic, renal, and neuromuscular responses, fully standardising these variables in free-living athletes is inherently challenging without adopting restrictive or residential protocols. Our study followed an ecological high-performance setting, where imposing strict control over daily routines would have compromised external validity and athlete compliance. Nevertheless, these factors remain recognised sources of interindividual variability, and their limited controllability represents an intrinsic constraint of real-world endurance-physiology research. This is particularly relevant for renal and muscle biomarkers, which are known to be sensitive to hydration status, workload, and thermal strain, as shown in recent field studies on exercise-induced kidney stress in endurance athletes [107].

Fourth, the cross-sectional design precludes inference about the stability of the phenotype over time or about responsiveness to training or recovery interventions. Longitudinal research is necessary to determine whether athletes shift between responder profiles across training phases or under varying load and recovery conditions [108].

Finally, an extended profiling model was also explored, including all post-exercise cardiovascular and renal marker variables. However, due to sample size limitations and increased dimensional overlap, many cyclists were classified as “mixed responders”, lacking a dominant physiological axis. Although a weighting algorithm partially disaggregated this heterogeneity, the added complexity limited practical applicability. Therefore, those exploratory heatmaps were discarded. Given the different approaches discussed in this manuscript, we propose selecting the simplest method with the greatest practical applicability in each case.

Future studies with larger samples may benefit from multivariate clustering or latent profile analysis to validate additional models, including more key variables. Future research should also validate responder profiles in more diverse groups, including female cyclists. It will help capture sex-dependent differences in neuromuscular, metabolic, and renal physiology. Incorporating real-time data from wearable sensors, such as HRV, continuous glucose monitoring, and muscle oxygenation, would improve ecological monitoring and allow high-resolution modelling of physiological changes during and after high-intensity efforts [109,110]. Machine-learning clustering, latent-profile analysis, and Bayesian network modelling could help quantify interactions across multiple domains at scale, as indicated by recent studies in endurance physiology.

On the other hand, intervention studies are also needed to test whether personalised strategies targeting domain-specific limitations such as neuromuscular resilience, glycolytic buffering capacity, or renal strain tolerance can shift athletes toward more adaptive responder profiles. Additionally, future protocols should incorporate structured dietary logs, hydration tracking, and sleep/HRV monitoring to reduce uncontrolled variability and strengthen interpretability of physiological interactions.

## 5. Conclusions

This study demonstrates that fatigue and recovery after a maximal 20-min cycling effort arise from coordinated metabolic, cardiovascular, neuromuscular, and renal responses that differ markedly between professional and amateur cyclists. Professionals showed higher FTP, more efficient cardiovascular and glycaemic recovery, and lower neuromuscular impairment, but substantial interindividual variability was evident in both groups.

To address this variability, we developed a four-variable profiling model (FTP, lactate, CK, and neuromuscular recovery) that successfully identified coherent physiological responder types. These profiles range from full systemic responders to metabolic-dominant or neuromuscular-limited athletes, providing an interpretable and operational framework that surpasses group-level comparisons and single-domain metrics.

Although extended biomarker models were explored, the focused four-variable approach proved the most practical and informative for application in trained cyclists. Future work should determine the stability of these responder profiles across training phases and assess whether targeted interventions can shift athletes toward more adaptive physiological patterns. Integrating such multidomain profiling into applied sport science may enhance individualised training prescription and athlete health monitoring in high-performance environments.

### Practical Applications

This study presents a framework for tailoring the monitoring and improvement of cycling performance, utilising a combination of field-based and laboratory-accessible tools. By analysing functional threshold power (FTP), neuromuscular performance (through Bosco tests), and selected biochemical markers like CK, lactate, glycaemia, and albuminuria, we can identify specific physiological responder profiles. These profiles, which range from full systemic responders to those focused on neuromuscular or metabolic aspects, can guide personalised training plans, enhance recovery methods, and help avoid excessive strain in both professional and amateur cyclists.

The suggested profiling model has practical benefits. It utilises non-invasive, repeatable tests that can be applied in high-performance or semi-professional settings, does not rely on costly equipment, and provides valuable insights into how individuals adapt differently. Coaches, sports scientists, and medical personnel can apply these profiles to customise training volumes, adjust pacing methods, and track recovery readiness. Additionally, the data support the use of simple field tests, such as SJ/CMJ, before and after FTP, as indicators of more complex physiological processes, making them suitable for ongoing monitoring outside of laboratory environments.

In practice, coaches could incorporate this four-variable model into a weekly or biweekly monitoring routine with minimal disruption. FTP can be reassessed every 4–6 weeks, while SJ and CMJ can be measured before and after key training sessions to monitor neuromuscular fatigue in real time. Lactate and CK can be selectively collected after high-intensity days or race simulations to identify disproportionate metabolic or muscular strain. Considering these variables together allows coaches to detect shifts toward glycolytic, neuromuscular, or systemic fatigue profiles. This enables timely adjustments in training load, session type, and recovery strategies, helping athletes stay within their adaptive zone and reducing the risk of accumulated fatigue or maladaptation.

## Figures and Tables

**Figure 1 jfmk-10-00472-f001:**
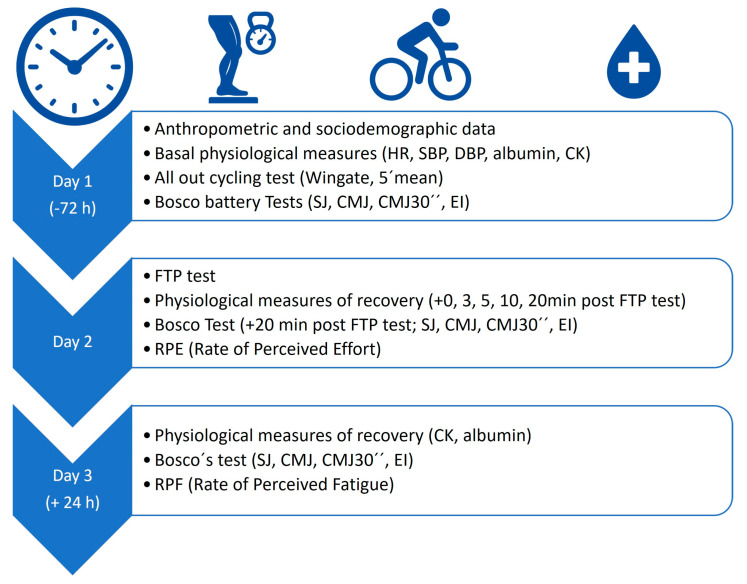
Study timeline and testing sequence. Overview of the three-day protocol including anthropometric and physiological assessments (Day 1), maximal cycling performance test (FTP test) with recovery measurements and neuromuscular evaluation (Day 2), and 24-h follow-up measurements (Day 3). Bosco Tests included squat jump (SJ), countermovement jump (CMJ), and countermovement jump with 30° flexion (CMJ30″). Physiological measures included heart rate (HR), systolic and diastolic blood pressure (SBP, DBP), glycaemia, temperature, creatine kinase (CK), and albuminuria. RPE: Rating of Perceived Effort; RPF: Rate of Perceived Fatigue.

**Figure 2 jfmk-10-00472-f002:**
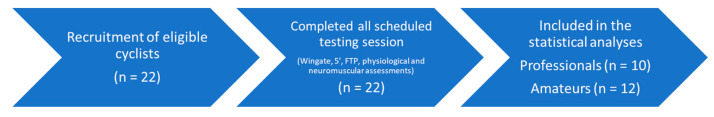
Study flowchart. Flow of participants throughout the study. Recruitment targeted cyclists who already met the eligibility criteria, resulting in 22 eligible participants. All 22 cyclists completed the scheduled testing sessions (Wingate, 5-min all-out test, 20-min FTP test, and physiological and neuromuscular assessments) and were included in the final statistical analyses.

**Figure 3 jfmk-10-00472-f003:**
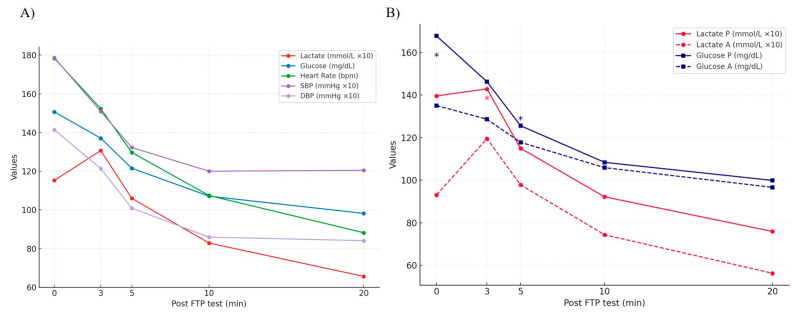
Dynamics of continuous variables immediately after the FTP test. The mean of each variable at each time point is represented simultaneously. The units used to measure each variable are indicated, and Lactate, SBP, and DBP values were multiplied by 10 to enhance visualisation. (**A**) Summary of the dynamics of all variables in all participants. (**B**) Glycaemia and Lactate dynamics in Professional (P) and Amateur (A) cyclists. The asterisk (*) denotes statistically significant differences (*p* < 0.05) between Professional (P) and Amateur (A) cyclists for the corresponding variable at the indicated time point.

**Figure 4 jfmk-10-00472-f004:**
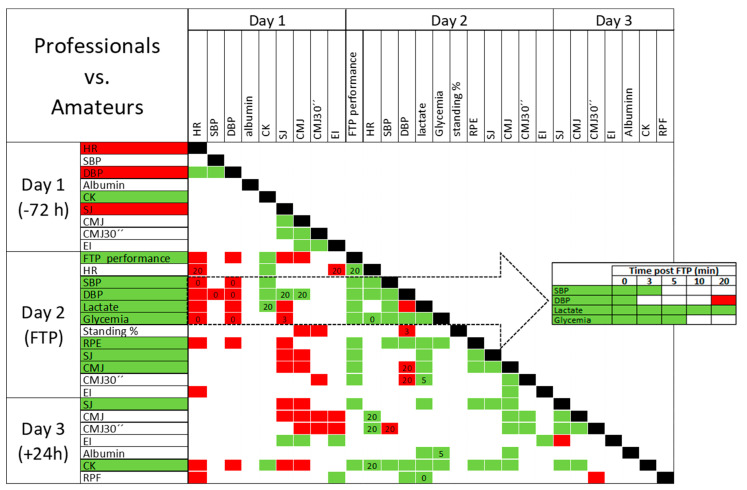
Physiology of cycling performance in Professional vs. Amateurs and how the variables interact. Left column: significant between-group differences (green: professionals > amateurs; red: amateurs > professionals). Matrix: significant correlations only (green: positive; red: negative), with numbers indicating post-FTP timepoints (min). Right panel: time-resolved group differences for SBP, DBP, lactate, and glycaemia (0–20 min; green/red: significant; white: non-significant).

**Figure 5 jfmk-10-00472-f005:**
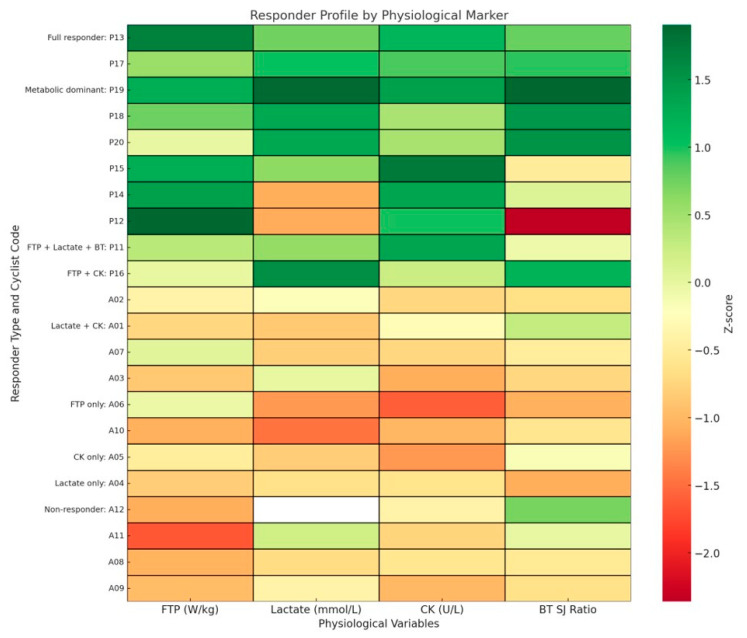
Heatmap of standardised Z-scores for four key physiological variables (FTP, post-exercise lactate, CK at 24 h, and squat jump BT ratio) across individual cyclists. Participants were classified into nine distinct physiological responder profiles based on validated thresholds for each variable. Profiles included: Full Responder, Metabolic Dominant, FTP + Lactate + BT, FTP + CK, Lactate + CK, FTP Only, CK Only, Lactate Only, and Non-Responder. Each row represents one cyclist, labelled with a P (professional) or A (amateur), grouped by their respective responder type.

**Table 1 jfmk-10-00472-t001:** Summary of main features and cycling performance of participants.

		Professional	Amateur	All Participants	*p*-Value
Main features	Age (years)	26.7 ± 7.2	28.4 ± 5.8	27.6 ± 6.4	0.540
	Height (cm)	178.9 ± 4.7	176.0 ± 5.9	177.3 ± 5.5	0.220
	Weight (kg)	63.9 ± 3.2	66.9 ± 4.4	65.5 ± 4.1	0.080
	Body mass index (kg/m^2^)	19.9 ± 0.7	21.6 ± 1.3	20.9 ± 1.3	0.001 *
20 min FTP test	Mean power (W)	347.4 ± 12.6	284.0 ± 20.0	312.8 ± 36.3	<0.001 *
	Relative mean power (W/kg)	5.4 ± 0.3	4.3 ± 0.4	4.8 ± 0.7	<0.001 *
	Standing time (%)	27.6 ± 15.1	27.3 ± 12.7	27.4 ± 13.5	0.960
Wingate test	Mean power (W)	826.7 ± 104.3	581.3 ± 91.9	692.9 ± 157.3	<0.001 *
	Relative mean power (W/kg)	12.9 ± 1.6	8.7 ± 1.2	10.6 ± 2.5	<0.001 *
	Peak power (W)	1052.5 ± 151.9	753.4 ± 97.7	889.3 ± 195.3	<0.001 *
5 min test	Mean power (W)	409.7 ± 28.3	345.7 ± 32.2	374.8 ± 44.1	<0.001 *
	Relative mean power (W/kg)	6.4 ± 0.5	5.2 ± 0.8	5.7 ± 0.8	<0.001 *

Anthropometric characteristics and cycling performance variables for professional and amateur cyclists. The table includes age, height, weight, BMI, and power outputs from the Wingate, 5-min all-out, and 20-min FTP tests. *p*-values correspond to Student’s *t*-tests comparing groups, and an asterisk indicates statistical significance (*p* < 0.05).

**Table 2 jfmk-10-00472-t002:** What features are related to professional status? And to better performance?

MEASURES		TIME						
		(DAY 1) PRE- FTP TEST			(DAY 2) POST-FTP TEST			(DAY 3) POST-FTP TEST
		−72 h	0 min	3 min	5 min	10 min	20 min	24 h
All out cycling test	30″ Peak	+C						
	30″ mean	+C						
	5’mean	+C, +P						
Bosco Test	SJ	−C	+C					+C
	CMJ	−C	+C					−C
	CMJ30″	−P	+C					−P
	EI	−P						−P
Perceived effort	RPE	+C, +A						
Physiological values	CK	+C						+C, +A
	HR	−C	+P				+C	
	SBP		+C, +P	+C, +P				
	DBP	−C	+C	+C			−C, −P	
	Glycaemia		+C	+C, +A	+A	+A		
	Lactate			+C	+C	+C, −P, +A	+C	

Features significantly associated (*p* < 0.05) with professional status are filled in green (positive association) or in red (negative association). Features significantly correlated (*p* < 0.05) to better performance in the FTP test are indicated with “+” if the correlation is positive and with “−“ is negative, when considering: “C” all participants, “A” only amateur group, and “P” only to the professional group.

**Table 3 jfmk-10-00472-t003:** How do variables interact?

	HR	SBP	DBP	LACTATE	GLYCAEMIA
HR					
SBP					
DBP					
LACTATE					
GLYCAEMIA					

A matrix to help us understand the biology of cycling performance. Significantly (*p* < 0.05) positive (green) and negative (red) correlations between variables are indicated.

## Data Availability

The original contributions presented in this study are included in the article. Further inquiries can be directed to the corresponding author.

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
