# Peer review of "Physiological Responder Profiles and Fatigue Dynamics in Prolonged Cycling"

_jfmk, 2025, doi:10.3390/jfmk10040472_

Round 1

Reviewer 1 Report

Comments and Suggestions for Authors

The manuscript entitled “The Limits of Performance During Prolonged Cycling Effort: Identification of Individual Physiological Responder Profiles Based on FTP, Neuromuscular Capacity, and Biomarker Dynamics” was reviewed. The article provides interesting information on the topic; however, adjustments need to be made so that the article can continue the path to publication.

I kindly ask if all changes made to the text be highlighted in yellow or a different color in the text.

Below are the reviewer's considerations to be adjusted in the manuscript.

Title:

  1. The title is too long and contains abbreviations. Please make it more concise and replace the abbreviations with full term, if necessary.

Abstract:

2- Since you have already started writing the abstract with the objectives, you can delete the word "Background".

3- Start the methods section by presenting the type of study and the sample profile. The beginning of the methods section seems to be part of the objectives. I suggest moving the beginning of the methods section to the objectives section and adjusting the text.

4- Present sociodemographic information of the participants and body composition, as well as the types of statistical tests used, and the minimum significance level adopted.

5- Mention in the results section the values ​​of the statistical tests, mean and standard deviation along with the variables analyzed.

6- The conclusions are too long; please make them more assertive, in order to answer the proposed objectives.

Introduction:

7- Explain in the introduction the physiological mechanisms that lead to fatigue in cycling. Take the opportunity to adjust the paragraph length, as some are too short. One possibility is to introduce new information or combine several paragraphs.

8- Present the current methods or strategies to mitigate or reduce fatigue in cycling.

9- Show in more detail the current state of the art on the subject, both in relation to professional and amateur athletes.

10- Adjust the objectives at the end of the introduction to match those presented in the abstract.

Methods:

11- Begin the description of the methods by presenting the type of study and the location of the data collection.

12- Describe the technical information used for anthropometric assessments, as well as the names of the manufacturers, brands, models, and sensitivity of the equipment.

13- It has been suggested that studies involving human subjects should have clinical trial registration. Did you register the research? If so, provide the number and access link.

14- Did you perform sample size calculations to determine if the number of people evaluated was sufficient?

15- Clarify the inclusion and exclusion criteria for participants. What minimum characteristics were used to classify a person as an amateur and a professional?

16- Develop a flowchart of the study so that the reader can understand the overall dynamics, from how many people were invited, how many were excluded, and how many were included in the statistical analyses.

17- Also, create a figure showing the intervention stages in relation to the timeline of events. Use real images from the collections or illustrative figures to create the artwork.

18- Describe the statistical analyses in chronological order. Start with the normality test, move on to the form of data presentation, mean and standard deviation or median and interquartile range (if the data are parametric or non-parametric, respectively). Address the statistical tests to be used and only at the end, present the information on the statistical programs used. There is no need to present the program's website.

Results:

19- Provide more characteristics of the participants in relation to body composition and sociodemographic profile.

20- In the text presenting Table 1, you should add more details about the main information observed.

21- Create legends for the tables explaining the abbreviations and statistical tests used. Regarding the p-value, you can standardize it with 3 decimal places after the comma. This will make the material more uniform. Apparently, you added legend information along with the table titles. Remove this information from there and place it below the table.

22- Transform Figure 2 into graphic figures so that the reader can better understand the results. For information that did not show significance, you can create a table and leave only the significant information as graphs.

Discussion:

23- Avoid repeating results in the discussion; focus on explaining the findings based on the literature and justify what happened in your research.

24- Bring in physiological elements to explain the findings.

25- Mention how your study contributed to the state of the art in the field.

26- ​​Describe the limitations of the study at the end of the discussion. Use the comments section to elaborate on the limitations.

Conclusions:

27- Adjust the conclusion to be more direct in answering the study's objectives. It's too long.

References:

28- There are many old references, please update to include more references from 2022 onwards.

Author Response

Dear reviewer,

We sincerely thank you for the time and effort dedicated to evaluating our manuscript, as well as for your constructive and insightful comments. Your suggestions have significantly strengthened the clarity, methodological soundness, and scientific contribution of the study. Below, we respond to each of your comments point by point, indicating the corresponding modifications made in the revised manuscript. All our responses are written in italics for clarity, and every change has been incorporated into the updated version of the manuscript.

Title:

  • The title is too long and contains abbreviations. Please make it more concise and replace the abbreviations with full term, if necessary.

 We appreciate the reviewer’s helpful suggestion. The title has been revised to be clearer, free of abbreviations, and better aligned with the study's main scientific contribution. The updated title is:

“Physiological Responder Profiles and Fatigue Dynamics in Prolonged Cycling.”

This revised version improves clarity, eliminates acronyms, and better aligns with the multidomain responder-profiling framework presented in the manuscript.

Abstract:

  • Since you have already started writing the abstract with the objectives, you can delete the word "Background".

The abstract now begins directly with the study’s objectives,

  • Start the methods section by presenting the type of study and the sample profile. The beginning of the methods section seems to be part of the objectives. I suggest moving the beginning of the methods section to the objectives section and adjusting the text.

The Methods subsection of the abstract has been updated to start with a clear explanation of the study design and sample profile. Repetitive introductory sentences that overlapped with the Objectives have been removed or moved to improve clarity and flow. The revised abstract now explicitly states that the study was observational and involved 22 trained male cyclists (10 professionals and 12 amateurs), followed by a brief description of the experimental protocol.

  • Present sociodemographic information of the participants and body composition, as well as the types of statistical tests used, and the minimum significance level adopted.

The abstract has been updated to include key sociodemographic details (age, height, weight) and the participants' competitive levels. A brief overview of the statistical analyses (t-tests, Spearman correlations) has been added, and the significance threshold (p < 0.05) is now clearly stated.

  • Mention in the results section the values ​​of the statistical tests, mean and standard deviation along with the variables analyzed.

The Results subsection of the abstract has been revised to include the main statistical results: means ± standard deviations, p-values for between-group comparisons (FTP, lactate, CK, neuromuscular recovery), and key correlation coefficients (FTP versus 5-minute all-out and Wingate power).

  • The conclusions are too long; please make them more assertive, in order to answer the proposed objectives.

The Conclusions subsection of the abstract has been rewritten to make it more concise, direct, and fully aligned with the stated objectives. The revised version is clearer and focuses on the key findings: interindividual variability, differences between professionals and amateurs, and the applied utility of the four-variable responder mode. The updated conclusion now reads:

“Fatigue and recovery in prolonged cycling show substantial interindividual variability across neuromuscular, metabolic, cardiovascular, and biochemical domains. Professional cyclists display faster recovery and more frequent multidomain responder profiles. The four-variable model (FTP, lactate, CK, SJ post/pre) enables clear identification of physiological responder types and offers a practical, integrative framework for personalised monitoring and performance management.”

Introduction:

  • Explain in the introduction the physiological mechanisms that lead to fatigue in cycling. Take the opportunity to adjust the paragraph length, as some are too short. One possibility is to introduce new information or combine several paragraphs.

In accordance with the reviewer’s suggestion, a new paragraph summarising the key physiological mechanisms underlying cycling fatigue has been added to the Introduction (between lines 51 and 60).

  • Present the current methods or strategies to mitigate or reduce fatigue in cycling.

A new paragraph has been added to the Introduction to summarize the main evidence-based strategies currently used to reduce fatigue in cycling, including pacing optimization, nutritional interventions (carbohydrate availability, hydration, and caffeine), strength and plyometric training, and heat-acclimation/cooling techniques (between lines 78 and 89).

  • Show in more detail the current state of the art on the subject, both in relation to professional and amateur athletes.

A new paragraph has been added to the Introduction (lines 90–101) in the revised manuscript summarizing the current state of the art regarding physiological, neuromuscular, and perceptual differences between professional and amateur cyclists.

  • Adjust the objectives at the end of the introduction to match those presented in the abstract.

The objectives in the Introduction (between lines 139-144) have been revised to ensure full alignment with those now presented in the Abstract (between lines 16-19).

Methods:

  • Begin the description of the methods by presenting the type of study and the location of the data collection.

We have revised the beginning of the Methods section to clearly specify the study design and the data collection setting. The revised paragraph has been added at the start of the “Materials and Methods” section (lines 149–154).

  • Describe the technical information used for anthropometric assessments, as well as the names of the manufacturers, brands, models, and sensitivity of the equipment.

We have included the technical specifications of the equipment used for anthropometric measurements. The scale (Seca 711) and stadiometer (Seca 220), including manufacturer, model, and measurement accuracy, are detailed in the Methods section (lines 190–194).

  • It has been suggested that studies involving human subjects should have clinical trial registration. Did you register the research? If so, provide the number and access link.

This study follows an observational, non-interventional design, involving standard physiological and performance assessments commonly used in sports science research. As no treatment or therapeutic procedure was allocated to participants, clinical trial registration was not applicable. The study was reviewed and approved by the Human Research Ethics Committee of the University of the Basque Country (M10-2021-191), conducted in accordance with the Declaration of Helsinki, and all participants provided written informed consent (lines 166-171).

  • Did you perform sample size calculations to determine if the number of people evaluated was sufficient?

Thank you for this interesting question. No a priori sample size calculation was performed because the study followed an observational, non-interventional physiological design focused on characterising multidomain responder profiles rather than estimating population effects or testing the efficacy of a predefined intervention. The sample size reflects all eligible professional and amateur cyclists available during the competitive period. Comparable sample sizes (n ≈ 10–30) have been commonly used in previous cycling physiology studies to examine neuromuscular, metabolic, and performance responses (Lucía et al., 2001; Jurasz et al., 2022).

  • Clarify the inclusion and exclusion criteria for participants. What minimum characteristics were used to classify a person as an amateur and a professional?

We have now clarified the inclusion and exclusion criteria and specified the criteria used to distinguish professional from amateur cyclists. The revised version explicitly defines eligibility (age range, training status, injury-free period, and exclusion conditions) and describes the competitive characteristics used to classify performance level. These additions have been incorporated into Section 2.2, lines 176–182 of the revised manuscript.

  • Develop a flowchart of the study so that the reader can understand the overall dynamics, from how many people were invited, how many were excluded, and how many were included in the statistical analyses.

A study flowchart has now been included to illustrate participant progression through the study. Because recruitment was targeted to cyclists who already met the eligibility criteria (competitive status, training load, and availability during the testing period), no invitations were sent to ineligible individuals, and no exclusions occurred after enrolment. All 22 eligible cyclists completed the full testing protocol and were included in the statistical analyses. The flowchart has been added as Figure 2 in the revised manuscript, and the remaining figures have been renumbered accordingly.

  • Also, create a figure showing the intervention stages in relation to the timeline of events. Use real images from the collections or illustrative figures to create the artwork.

A new figure illustrating the full timeline of the study procedures has been created and included as Figure 1. The graphic summarises all testing stages (anthropometry, physiological measurements, cycling tests, Bosco tests, and recovery assessments) across the three days of data collection.

  • Describe the statistical analyses in chronological order. Start with the normality test, move on to the form of data presentation, mean and standard deviation or median and interquartile range (if the data are parametric or non-parametric, respectively). Address the statistical tests to be used and only at the end, present the information on the statistical programs used. There is no need to present the program's website.

In the revised manuscript, the Statistical Analysis section has been fully reorganised to follow the requested chronological order. The updated version now begins with the assessment of normality, then specifies the criteria for data presentation (mean ± SD), followed by the analytical procedures used for group comparisons (t-tests, mixed ANOVA), correlation analyses, and regression modelling of recovery dynamics. Only after describing the statistical workflow do we present the procedures for the heatmap and responder-threshold classification. Finally, the software information appears at the end of the section, without URLs, as recommended.

Results:

  • Provide more characteristics of the participants in relation to body composition and sociodemographic profile.

Additional participant characteristics have now been incorporated as requested. Specifically, body mass index (BMI) has been added. In Materials and Methods, BMI calculation was included in the anthropometric procedures (lines 194–195). In the Results section, BMI group comparison and statistical outcome were added to the text (line 302) and incorporated into Table 1. And in the Discussion, the relevance of BMI differences between professional and amateur cyclists has been integrated into the interpretation of group differences (lines 392–395). These additions provide a more complete sociodemographic and anthropometric description of the sample while maintaining methodological coherence.

  • In the text presenting Table 1, you should add more details about the main information observed.

We have expanded the caption of Table 1 to provide clearer information about the variables included and the statistical comparisons presented, without repeating the descriptive results already provided in the text (lines 304-308).

  • Create legends for the tables explaining the abbreviations and statistical tests used. Regarding the p-value, you can standardize it with 3 decimal places after the comma. This will make the material more uniform. Apparently, you added legend information along with the table titles. Remove this information from there and place it below the table.

The legends for both tables have now been revised, and all p-values have been standardised to three decimal places to improve consistency throughout the manuscript. The explanatory information previously displayed in the table titles has been removed and relocated to the legend area beneath each table, as requested.

  • Transform Figure 2 into graphic figures so that the reader can better understand the results. For information that did not show significance, you can create a table and leave only the significant information as graphs.

Following your recommendation, we substantially improved the presentation of the physiological results:

We have improved the legend of Figure 3 to indicate the specific values for each variable at each time, and the asterisk (*) denotes statistically significant differences (p < 0.05) between Professional (P) and Amateur (A) cyclists for the corresponding variable at the indicated time point (between lines 351-355).

To isolate only statistically significant findings, as suggested, we created an additional new figure. This is now included as Figure 4 (between lines 356-373), a combined interaction matrix + time-resolved significance panel. The matrix displays only significant correlations between physiological variables (green = positive; red = negative). The left column highlights only significant between-group differences (professionals vs amateurs). The panel on the right shows only the significant post-FTP differences across time points (0–20 min) for SBP, DBP, lactate, and glycaemia, exactly as requested. All non-significant comparisons (p ≥ 0.05) appear as blank cells.

Together, Figures 3 and 4 now provide a much clearer, more focused visualisation of the physiological responses, separating full dynamic information from strictly significant results. This dual approach aligns precisely with your recommendation and greatly improves readability and interpretability.

 Discussion:

  • Avoid repeating results in the discussion; focus on explaining the findings based on the literature and justify what happened in your research.

In the revised manuscript, the Discussion has been substantially restructured to avoid repeating numerical results and to focus instead on physiological interpretation and literature-based justification. Across lines 347–564 of the revised version, descriptive metrics previously duplicated from the Results (e.g., lactate values, albuminuria magnitudes, CK ranges, jump-test differences, and subgroup-specific numbers) have been removed. These sections now highlight the mechanistic interpretation of metabolic–renal stress, cardiovascular recovery, neuromuscular adaptation, and perception of effort. Similarly, we discuss how our findings fit within current evidence on endurance physiology, and how the identified responder profiles provide an integrative explanation for the observed patterns. This restructuring ensures that the Discussion explains why the results occurred rather than merely reiterating what the results were, thereby meeting the reviewer’s request to focus on interpretation and scientific justification.

  • Bring in physiological elements to explain the findings.

In the revised manuscript, we have substantially strengthened the physiological interpretation throughout the Discussion. Rather than simply describing patterns, we now explain them using established mechanisms from cardiovascular, metabolic, neuromuscular, and renal physiology. Specifically, we incorporated mechanistic explanations for: 1) glycolytic and acid–base dynamics underlying high lactate responses and their link to glomerular permeability and albuminuria; 2) neuromuscular fatigue mechanisms, and cardiovascular adaptations that differentiate professional from amateur; 3) integrated metabolic–renal responses, defining the metabolic–renal responder phenotype; 4) perceptual–effort regulation.

  • Mention how your study contributed to the state of the art in the field.

We have clarified in the revised Discussion how the study advances the current state of the art. The text now highlights that traditional cycling metrics (e.g., FTP, VO₂max, isolated metabolic or neuromuscular markers) cannot capture the multisystem interactions that shape fatigue, recovery, and performance. Our multidomain responder-profile model integrating FTP, lactate, CK, and neuromuscular recovery provides a more comprehensive characterisation of athlete physiology and reveals coordinated metabolic, cardiovascular, renal, and neuromuscular patterns that single-domain approaches overlook. This point is now explicitly stated in the revised Discussion, highlighting the study's conceptual and applied contributions.

  • Describe the limitations of the study at the end of the discussion. Use the comments section to elaborate on the limitations.

In accordance with the recommendation, we have added a dedicated “Limitations and future directions” section at the end of the Discussion (lines 658–710). This new section summarises all major methodological constraints of the study, including sample size, male-only cohort, self-paced testing, and uncontrollable real-world physiological confounders. Likewise, it expands on their implications for interpretation. The section also integrates a brief rationale for the discarded extended profiling model and outlines the main steps required for validation in future research.

 Conclusions:

  • Adjust the conclusion to be more direct in answering the study's objectives. It's too long.

In the revised manuscript, we have rewritten the conclusion to be more direct, concise, and clearly aligned with the study’s objectives. The updated section (lines 698–716) now emphasizes key findings regarding how professionals and amateurs differ, how the four-variable responder-profiling model improves on traditional group comparisons, and what this framework adds to applied performance analysis.

 References:

  • There are many old references, please update to include more references from 2022 onwards.

In the revised manuscript, we have updated the bibliography by incorporating multiple recent references (2022–2024) across the Introduction, Methods, Discussion, and Limitations sections. These include current evidence on cycling physiology, fatigue mechanisms, renal and metabolic responses to endurance exercise, sex differences, wearable-derived monitoring, and machine-learning approaches for athlete profiling. The reference list now reflects a substantially more contemporary body of literature, thereby strengthening the study's s

Reviewer 2 Report

Comments and Suggestions for Authors

The manuscript is generally clear, the methodology is solid, and the results are presented with enough detail. However, a few sections could be improved for readability, clarity, and scientific depth. Below I provide specific comments.

  • The idea of “physiological responder profiles” is appealing, but the introduction could better highlight how this model advances beyond traditional FTP or VO₂max-based analysis. Right now, the rationale feels slightly diluted among citations. Consider emphasizing early on why multidomain profiling (FTP + BT + biomarkers) fills a gap in current cycling science.
  • The procedures are described carefully, but several methodological choices need clearer justification. For example:Why exactly were thresholds (FTP ≥ 5.5 W/kg, lactate > 13 mmol/L, CK > 400 U/L, SJ ratio ≥ 0.90) chosen?

  • Were they based on prior empirical cut-offs or internal distributions?The use of a small, all-male sample (n = 22) should be discussed more explicitly as a limitation.

  • In my opinion, adding a simple figure or flow diagram of the experimental protocol (days 1–3) would make the design easier to follow.

  • The analysis seems appropriate, but the authors rely mainly on t-tests and correlations. Given the repeated-measures structure (pre/post/24 h), a mixed ANOVA or linear mixed model could provide stronger evidence. 

  • The description of regression analyses for recovery dynamics is clear, but the presentation of results could be more concise (perhaps summarized in one composite figure).

  • The results are rich and comprehensive, but some sections could be streamlined. For instance, the paragraph on albuminuria and renal stress, while interesting, slightly diverges from the main aim unless connected back to the responder profiles.

  • It might help to summarise the key physiological patterns in one table showing which domains were dominant in professionals vs. amateurs.

  • The discussion sometimes repeats results. Try to synthesize instead of restating numerical values.

  • The “Practical Applications” section is excellent. To strengthen it, add one short paragraph about how coaches could actually implement this four-variable model in weekly monitoring (e.g., when to test, how to interpret).

  • A dedicated paragraph summarizing main limitations (sample size, lack of female participants, self-paced effort variability, possible confounders like nutrition and hydration) would improve transparency.

  • Future directions could include validation in a larger cohort and integration with wearable sensors or machine-learning clustering methods.

  • Minor Comments

  • Some abbreviations (e.g., RPF, EI) appear before being defined; ensure each is defined at first mention.

  • Minor English corrections would improve flow — e.g., “the factor of threshold power” → “functional threshold power,” “physiological value predictor” → “physiological performance indicator.”

  • Figures 1–2 captions could include brief interpretations (“higher FTP associated with…”).

  • Reference formatting appears consistent, but ensure all DOIs are included.

Overall, this is a strong and promising manuscript. The combination of physiological, neuromuscular, and biochemical data offers an integrative view rarely seen in small applied-sport studies. 

Author Response

Dear Reviewer,

We are grateful for your careful evaluation of our manuscript and for the thoughtful, detailed feedback you provided. Your comments have been highly valuable in refining the structure, strengthening the methodological rigor, and improving the overall clarity of the work. In the following section, we address each of your points individually, specifying the corresponding revisions made throughout the manuscript. All responses are presented in italics for ease of reading, and every suggested adjustment has been incorporated into the revised version.

Comments and Suggestions for Authors

The manuscript is generally clear, the methodology is solid, and the results are presented with enough detail. However, a few sections could be improved for readability, clarity, and scientific depth. Below I provide specific comments.

  • The idea of “physiological responder profiles” is appealing, but the introduction could better highlight how this model advances beyond traditional FTP or VO₂max-based analysis. Right now, the rationale feels slightly diluted among citations. Consider emphasizing early on why multidomain profiling (FTP + BT + biomarkers) fills a gap in current cycling science.

We have strengthened the conceptual rationale for moving beyond single-domain metrics such as FTP or VO₂max. The revised text now explicitly clarifies how a multidomain responder-profiling framework meaningfully advances current analytical approaches in cycling science. In particular, the Introduction now provides a clearer theoretical foundation for the model by first outlining the multisystem physiological mechanisms that drive cycling fatigue (lines 52–61), emphasising that traditional aerobic or power-based indicators cannot capture the full spectrum of performance-limiting processes. We then summarise the main evidence-based strategies for mitigating fatigue (lines 79–90), illustrating that cycling performance arises from the interaction among metabolic, cardiovascular, neuromuscular, and perceptual domains rather than from any single determinant. This is followed by an updated synthesis of known physiological, neuromechanical, and perceptual differences between professional and amateur cyclists (lines 91–102), highlighting that these distinctions already span multiple systems. Finally, we articulate more explicitly the need for multidomain profiling (lines 103–119), explaining that combining FTP with neuromuscular measures (Bosco test), metabolic markers (lactate, glucose), and renal–muscle biomarkers (albuminuria, CK) allows a more comprehensive characterisation of individual physiological regulation than FTP or VO₂max alone.

  • The procedures are described carefully, but several methodological choices need clearer justification. For example:Why exactly were thresholds (FTP ≥ 5.5 W/kg, lactate > 13 mmol/L, CK > 400 U/L, SJ ratio ≥ 0.90) chosen? Were they based on prior empirical cut-offs or internal distributions? The use of a small, all-male sample (n = 22) should be discussed more explicitly as a limitation.

In the revised manuscript, we now provide a clear justification for each responder threshold. Specifically, we explain that the cut-offs were not arbitrarily defined nor derived from our internal distributions, but are based on prior empirical evidence describing physiologically meaningful boundaries in endurance athletes. As detailed in lines 617–634, an FTP ≥ 5.5 W·kg¹ reflects the minimum value commonly reported for professional-level endurance performance; lactate > 13 mmol·L¹ corresponds to high glycolytic activation observed in maximal cycling protocols; CK > 400 U·L¹ is a recognized marker of significant exercise-induced muscle membrane stress; and an SJ post/pre ratio 0.90 is widely used to indicate meaningful neuromuscular fatigue and incomplete SSC recovery. These additions clarify that the thresholds are physiologically grounded, supported by prior literature, and chosen to represent distinct, mechanistically coherent functional domains.

Regarding the comment “The use of a small, all-male sample (n = 22) should be discussed more explicitly as a limitation,” this issue has now been clearly addressed in the newly added Limitations and Future Directions section (lines 658–667). We discuss (i) the small sample size, (ii) the inclusion of only trained male cyclists, and (iii) the resulting limitations on generalizability and on examining sex-specific neuromuscular, metabolic, and renal responses. Appropriate references have also been added to support the importance of sex-based differences in fatigability and recovery.

  • In my opinion, adding a simple figure or flow diagram of the experimental protocol (days 1–3) would make the design easier to follow.

We have now added a dedicated figure illustrating the three-day experimental protocol in a clear, chronological flow format. This figure (now Figure 1) summarizes all assessments conducted each day, including anthropometric measurements, physiological tests, neuromuscular evaluations, and the timing of post-FTP measurements. It enhances the readability and transparency of the study design. This addition is shown in Figure 1 of the revised version.

  • The analysis seems appropriate, but the authors rely mainly on t-tests and correlations. Given the repeated-measures structure (pre/post/24 h), a mixed ANOVA or linear mixed model could provide stronger evidence.

We have re-analyzed all variables suitable for parametric testing using one-way ANOVA, as described in the materials and methods. The results confirmed that the pattern of significant differences between professional and amateur cyclists was consistent when comparing Student’s t-tests and ANOVA results. This consistency enhances the robustness of the group-level comparisons and supports the validity of the analytical method used in the study. Thank you for the advice.

  • The description of regression analyses for recovery dynamics is clear, but the presentation of results could be more concise (perhaps summarized in one composite figure).

We have streamlined the presentation of recovery-dynamics results by integrating them into a single composite visualisation. Specifically, the new Figure 4 now summarises: (i) the significant group differences across time points, (ii) the correlation matrix of key metabolic, cardiovascular and neuromuscular variables, and (iii) the temporal evolution of SBP, DBP, lactate and glycaemia during the 0–20 min recovery window.

  • The results are rich and comprehensive, but some sections could be streamlined. For instance, the paragraph on albuminuria and renal stress, while interesting, slightly diverges from the main aim unless connected back to the responder profiles.

We revised the section on albuminuria and renal stress to streamline the narrative and ensure it is clearly integrated into the study’s multidomain responder-profiling framework. Specifically, we removed redundant physiological descriptions, synthesized the mechanistic content, and added a direct conceptual link explaining how albuminuria contributes to the metabolic–renal responder phenotype. This change improves clarity and aligns with the study's main goal. The revised text now appears in lines 515–544.

  • It might help to summarise the key physiological patterns in one table showing which domains were dominant in professionals vs. amateurs.

In the revised manuscript, we included an integrated visual summary, now shown as Figure 4, as commented previously. It offers a more comprehensive multidomain overview than a standard table. This enhances the clarity of the physiological structure of the results.

  • The discussion sometimes repeats results. Try to synthesize instead of restating numerical values.

In the revised discussion, we have streamlined the narrative and removed redundant numerical detail, especially in sections on albuminuria, CK responses, recovery dynamics, and professional–amateur comparisons. Instead of restating values already presented in the Results, the text now focuses on interpretation, mechanistic integration, and phenotype-level implications. These revisions substantially reduce repetition and improve conceptual flow while preserving analytical depth.

  • The “Practical Applications” section is excellent. To strengthen it, add one short paragraph about how coaches could actually implement this four-variable model in weekly monitoring (e.g., when to test, how to interpret).

we have added a concise paragraph at the end of the Practical Applications section that explains how coaches can operationalise the four-variable model in weekly monitoring routines (lines 750–758). The new text outlines practical testing frequency (FTP every 4–6 weeks; SJ/CMJ pre/post key sessions; lactate and CK after high-intensity workloads) and clarifies how these variables can be jointly interpreted to identify shifts toward glycolytic, neuromuscular, or systemic-fatigue profiles.

  • A dedicated paragraph summarizing main limitations (sample size, lack of female participants, self-paced effort variability, possible confounders like nutrition and hydration) would improve transparency.

A complete paragraph addressing the main limitations (sample size, male-only cohort, self-paced protocol, and real-world confounders) has been incorporated in the new section (lines 658–694).

  • Future directions could include validation in a larger cohort and integration with wearable sensors or machine-learning clustering methods.

Future research directions (validation in larger cohorts, inclusion of females, integration of wearable sensors, machine-learning clustering, and targeted interventions) have been added to the same section (lines 695–710).

Minor Comments

  • Some abbreviations (e.g., RPF, EI) appear before being defined; ensure each is defined at first mention.

We have now carefully reviewed the entire manuscript and corrected the instances where abbreviations appeared before being introduced

  • Minor English corrections would improve flow — e.g., “the factor of threshold power” → “functional threshold power,” “physiological value predictor” → “physiological performance indicator.”

All suggested English refinements have now been implemented.

  • Figures 1–2 captions could include brief interpretations (“higher FTP associated with…”).

In the revised manuscript, we added a brief interpretative sentence to the captions of the figures corresponding to the original Figures 1–2 (now Figures 3 and 5). The caption of Figure 3 has been expanded to include a brief interpretative statement summarising the main post-FTP physiological dynamics and group differences. In the case of Figure 5, the following sentence has been added “This representation highlights that professional cyclists tended to cluster in profiles with multi-domain physiological responsiveness, whereas amateurs more often showed single-domain or non-responder patterns.”.  These short statements clarify the physiological meaning of the visual patterns while avoiding redundancy with the Results section.

  • Reference formatting appears consistent, but ensure all DOIs are included.

  • Overall, this is a strong and promising manuscript. The combination of physiological, neuromuscular, and biochemical data offers an integrative view rarely seen in small applied-sport studies.

Round 2

Reviewer 1 Report

Comments and Suggestions for Authors

Dear authors,

Thank you for providing the revised manuscript. After careful reading, it has been found that the text has improved even further in quality. Therefore, my suggestion is that the article be accepted for publication.

Reviewer 2 Report

Comments and Suggestions for Authors

I confirm that all of my comments have been thoroughly and appropriately addressed in the revised manuscript. The authors have provided clear, substantive responses and implemented all necessary corrections, clarifications, and improvements. The revised version is substantially strengthened in terms of methodological transparency, interpretative depth, and overall readability.

I recommend the manuscript for acceptance.